# Bayesian Kernelized Tensor Factorization as Surrogate for Bayesian Optimization

## Abstract

Bayesian optimization (BO) primarily uses Gaussian processes (GP) as the key surrogate model, mostly with a simple stationary and separable kernel function such as the squared-exponential kernel with automatic relevance determination (SE-ARD). However, such simple kernel specifications are deficient in learning functions with complex features, such as being nonstationary, nonseparable, and multimodal. Approximating such functions using a local GP, even in a low-dimensional space, requires a large number of samples, not to mention in a high-dimensional setting. In this paper, we propose to use Bayesian Kernelized Tensor Factorization (BKTF)—as a new surrogate model—for BO in a $D$-dimensional Cartesian product space. Our key idea is to approximate the underlying $D$-dimensional solid with a fully Bayesian low-rank tensor CP decomposition, in which we place GP priors on the latent basis functions for each dimension to encode local consistency and smoothness. With this formulation, information from each sample can be shared not only with neighbors but also across dimensions. Although BKTF no longer has an analytical posterior, we can still efficiently approximate the posterior distribution through Markov chain Monte Carlo (MCMC) and obtain prediction and full uncertainty quantification (UQ). We conduct numerical experiments on both standard BO test functions and machine learning hyperparameter tuning problems, and our results show that BKTF offers a flexible and highly effective approach for characterizing complex functions with UQ, especially in cases where the initial sample size and budget are severely limited.

## 1 Introduction

For many applications in sciences and engineering, such as emulation-based studies, design of experiments, and automated machine learning, the goal is to optimize a complex black-box function $f(\boldsymbol{x})$ in a $D$-dimensional space, for which we have limited prior knowledge. The main challenge in such optimization problems is that we aim to efficiently find global optima rather than local optima, while the objective function $f$ is often gradient-free, multimodal, and computationally expensive to evaluate. Bayesian optimization (BO) offers a powerful statistical approach to these problems, particularly when the observation budgets are limited [1, 2, 3]. A typical BO framework consists of two components to balance exploitation and exploration: the surrogate and the acquisition function (AF). The surrogate is a probabilistic model that allows us to estimate $f(\boldsymbol{x})$ with uncertainty at a new location $\boldsymbol{x}$, and the AF is used to determine which location to query next.

Gaussian process (GP) regression is the most widely used surrogate for BO [3, 4], thanks to its appealing properties in providing analytical derivations and uncertainty quantification (UQ). The choice of kernel/covariance function is a critical decision in GP models; for multidimensional BO problems, perhaps the most popular kernel is the ARD (automatic relevance determination)—

Squared-Exponential (SE) or Matérn kernel [4]. Although this specification has certain numerical advantages and can help automatically learn the importance of input variables, a key limitation is that it implies/assumes that the underlying stochastic process is both stationary and separable, and the value of the covariance function between two random points quickly goes to zero with the increase of input dimensionality. These assumptions can be problematic for complex real-world processes that are nonstationary and nonseparable, as estimating the underlying function with a simple ARD kernel would require a large number of observations. A potential solution to address this issue is to use more flexible kernel structures. The additive kernel, for example, is designed to characterize a more "global" and nonstationary structure by restricting variable interactions [5], and it has demonstrated great success in solving high-dimensional BO problems (see, e.g., [6, 7, 8]). However, in practice using additive kernels requires strong prior knowledge to determine the proper interactions and involves many kernel hyperparameters to learn [9]. Another emerging solution is to use deep GP [10], such as in [11]; however, for complex multidimensional functions, learning a deep GP model will require a large number of samples.

In this paper, we propose to use *Bayesian Kernelized Tensor Factorization* (BKTF) [12, 13, 14] as a flexible and adaptive surrogate model for BO in a $D$-dimensional Cartesian product space. BKTF is initially developed for modeling multidimensional spatiotemporal data with UQ, for tasks such as spatiotemporal kriging/cokriging. This paper adapts BKTF to the BO setting, and our key idea is to characterize the multivariate objective function $f(\boldsymbol{x}) = f(x_1, \ldots, x_D)$ for a specific BO problem using low-rank tensor CANDECOMP/PARAFAC (CP) factorization with random basis functions. Unlike other basis-function models that rely on known/deterministic basis functions [15], BKTF uses a hierarchical Bayesian framework to achieve high-quality UQ in a more flexible way—GP priors are used to model the basis functions, and hyperpriors are used to model kernel hyperparameters in particular for the lengthscale that characterizes the scale of variation.

Figure 1 shows the comparison between BKTF and GP surrogates when optimizing a 2D function that is nonstationary, nonseparable, and multimodal. The details of this function and the BO experiments are provided in Appendix 7.3, and related code is given in Supplementary material. For this case, GP becomes ineffective in finding the global solution, while BKTF offers superior flexibility and adaptability to characterize the multidimensional process from limited data. Different from GP-based surrogate models, BKTF no longer has an analytical posterior; however, efficient inference and acquisition can be achieved through Markov chain Monte Carlo (MCMC) in an element-wise learning way, in which we update basis functions and kernel hyperparameters using Gibbs sampling and slice sampling respectively [14]. For the optimization, we first use MCMC samples to approximate the posterior distribution of the whole tensor and then naturally define the upper confidence bound (UCB) as AF. This process is feasible for many real-world applications that can be studied in a discretized tensor product space, such as experimental design and automatic machine learning (ML). We conduct extensive experiments on both standard optimization and ML hyperparameter tuning tasks. Our results show that BKTF achieves a fast global search for optimizing complex objective functions under limited initial data and observation budgets.

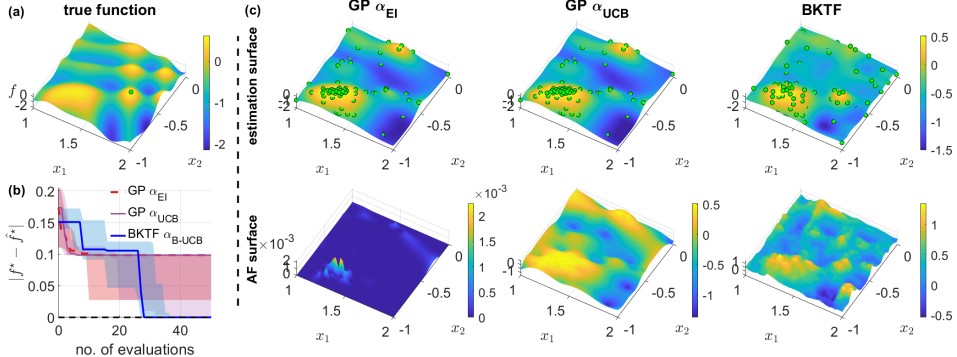

Figure 1: BO for a 2D nonstationary nonseparable function: (a) True function surface, where the global maximum is marked; (b) Comparison between BO models using GP surrogates (with two AFs) and BKTF with 30 random initial observations, averaged over 20 replications; (c) Specific results of one run, including the final mean surface for $f$, in which green dots denote the locations of selected candidates, and the corresponding AF surface.

## 2 Preliminaries

Throughout this paper, we use lowercase letters to denote scalars, e.g., $x$, boldface lowercase letters to denote vectors, e.g., $\boldsymbol{x} = (x_1, \ldots, x_D)^\top \in \mathbb{R}^D$, and boldface uppercase letters to denote matrices, e.g., $\boldsymbol{X} \in \mathbb{R}^{M \times N}$. For a matrix $\boldsymbol{X}$, we denote its determinant by $\det(\boldsymbol{X})$. We use $\boldsymbol{I}_N$ to represent an identity matrix of size $N$. Given two matrices $\boldsymbol{A} \in \mathbb{R}^{M \times N}$ and $\boldsymbol{B} \in \mathbb{R}^{P \times Q}$, the Kronecker product is defined as $\boldsymbol{A} \otimes \boldsymbol{B} = \begin{bmatrix} a_{1,1}\boldsymbol{B} & \cdots & a_{1,N}\boldsymbol{B} \\ \vdots & \ddots & \vdots \\ a_{M,1}\boldsymbol{B} & \cdots & a_{M,N}\boldsymbol{B} \end{bmatrix} \in \mathbb{R}^{MP \times NQ}$. The outer product of two vectors $\boldsymbol{a}$ and $\boldsymbol{b}$ is denoted by $\boldsymbol{a} \circ \boldsymbol{b}$. The vectorization operation $\text{vec}(\boldsymbol{X})$ stacks all column vectors in $\boldsymbol{X}$ as a single vector. Following the tensor notation in [16], we denote a third-order tensor by $\mathcal{X} \in \mathbb{R}^{M \times N \times P}$ and its mode-$k$ ($k = 1, 2, 3$) unfolding by $\boldsymbol{X}_{(k)}$, which maps a tensor into a matrix. Higher-order tensors can be defined in a similar way.

Let $f : \mathcal{X} = \mathcal{X}_1 \times \ldots \times \mathcal{X}_D \to \mathbb{R}$ be a black-box function that could be nonconvex, derivative-free, and expensive to evaluate. BO aims to address the global optimization problem:

$$\boldsymbol{x}^\star = \arg\max_{\boldsymbol{x} \in \mathcal{X}} f(\boldsymbol{x}), \quad f^\star = \max_{\boldsymbol{x} \in \mathcal{X}} f(\boldsymbol{x}) = f(\boldsymbol{x}^\star). \tag{1}$$

BO solves this problem by first building a probabilistic model for $f(\boldsymbol{x})$ (i.e., surrogate model) based on initial observations and then using the model to decide where in $\mathcal{X}$ to evaluate/query next. The overall goal of BO is to find the global optimum of the objective function through as few evaluations as possible. Most BO models rely on a GP prior for $f(\boldsymbol{x})$ to achieve prediction and UQ:

$$f(\boldsymbol{x}) = f(x_1, x_2, \ldots, x_D) \sim \mathcal{GP}\left(m(\boldsymbol{x}), k(\boldsymbol{x}, \boldsymbol{x}')\right), \ x_d \in \mathcal{X}_d, \ d = 1, \ldots, D, \tag{2}$$

where $k$ is a valid kernel/covariance function and $m$ is a mean function that can be generally assumed to be 0. Given a finite set of observation points $\{\boldsymbol{x}_i\}_{i=1}^n$ with $\boldsymbol{x}_i = \left(x_1^i, \ldots, x_D^i\right)^\top$, the vector of function values $\boldsymbol{f} = (f(\boldsymbol{x}_1), \ldots, f(\boldsymbol{x}_n))^\top$ has a multivariate Gaussian distribution $\boldsymbol{f} \sim \mathcal{N}(\boldsymbol{0}, \boldsymbol{K})$, where $\boldsymbol{K}$ denotes the $n \times n$ covariance matrix. For a set of observed data $\mathcal{D} = \{\boldsymbol{x}_i, y_i\}_{i=1}^n$ with i.i.d. Gaussian noise, i.e., $y_i = f(\boldsymbol{x}_i) + \epsilon_i$ where $\epsilon_i \sim \mathcal{N}(0, \tau^{-1})$, GP gives an analytical posterior distribution of $f(\boldsymbol{x})$ at an unobserved point $\boldsymbol{x}^*$:

$$f(\boldsymbol{x}^*) \mid \mathcal{D}_n \sim \mathcal{N}\left(\boldsymbol{k}_{\boldsymbol{x}^* \boldsymbol{X}}\left(\boldsymbol{K} + \tau^{-1}\boldsymbol{I}_n\right)^{-1}\boldsymbol{y}, \ k_{\boldsymbol{x}^* \boldsymbol{x}^*} - \boldsymbol{k}_{\boldsymbol{x}^* \boldsymbol{X}}\left(\boldsymbol{K} + \tau^{-1}\boldsymbol{I}_n\right)^{-1}\boldsymbol{k}_{\boldsymbol{x}^* \boldsymbol{X}}^\top\right), \tag{3}$$

where $k_{\boldsymbol{x}^* \boldsymbol{x}^*}, \boldsymbol{k}_{\boldsymbol{x}^* \boldsymbol{X}} \in \mathbb{R}^{1 \times n}$ are variance of $\boldsymbol{x}^*$, covariances between $\boldsymbol{x}^*$ and $\{\boldsymbol{x}_i\}_{i=1}^n$, respectively, and $\boldsymbol{y} = (y_1, \ldots, y_n)^\top$.

Based on the posterior distributions of $f$, one can compute an AF, denoted by $\alpha : \mathcal{X} \to \mathbb{R}$, for a new candidate $\boldsymbol{x}^*$ and evaluate how promising $\boldsymbol{x}^*$ is. In BO, the next query point is often determined by maximizing a selected/predefined AF, i.e., $\boldsymbol{x}_{n+1} = \arg\max_{\boldsymbol{x} \in \mathcal{X}} \alpha(\boldsymbol{x} \mid \mathcal{D}_n)$. Most AFs are built on the predictive mean and variance; for example, a commonly used AF is the **expected improvement** (EI) [1]:

---

**Algorithm 1:** Basic BO process

**Input:** Initial dataset $\mathcal{D}_0$ and a trained surrogate model; total budget $N$.

**for** $n = 1, \ldots, N$ **do**

    Compute the posterior distribution of $f$ using all available data;

    Find next evaluation point $\boldsymbol{x}_n \in \mathbb{R}^D$ by optimizing the AF;

    Augment data $\mathcal{D}_n = \mathcal{D}_{n-1} \cup \{\boldsymbol{x}_n, y_n\}$, update surrogate model.

---

$$\alpha_{\text{EI}}(\boldsymbol{x} \mid \mathcal{D}_n) = \sigma(\boldsymbol{x})\varphi\left(\frac{\Delta(\boldsymbol{x})}{\sigma(\boldsymbol{x})}\right) + |\Delta(\boldsymbol{x})|\Phi\left(\frac{\Delta(\boldsymbol{x})}{\sigma(\boldsymbol{x})}\right), \tag{4}$$

where $\Delta(\boldsymbol{x}) = \mu(\boldsymbol{x}) - f_n^\star$ is the expected difference between the proposed point $\boldsymbol{x}$ and the current best solution, $f_n^\star = \max_{\boldsymbol{x} \in \{\boldsymbol{x}_i\}_{i=1}^n} f(\boldsymbol{x})$ denotes the best function value obtained so far; $\mu(\boldsymbol{x})$ and $\sigma(\boldsymbol{x})$ are the predictive mean and predictive standard deviation at $\boldsymbol{x}$, respectively; and $\varphi(\cdot)$ and $\Phi(\cdot)$ denote the probability density function (PDF) and the cumulative distribution function (CDF) of standard normal, respectively. Another widely applied AF for maximization problems is the **upper confidence bound** (UCB) [17]:

$$\alpha_{\text{UCB}}(\boldsymbol{x} \mid \mathcal{D}_n, \beta) = \mu(\boldsymbol{x}) + \beta\sigma(\boldsymbol{x}), \tag{5}$$

where $\beta$ is a tunable parameter that balances exploration and exploitation. The general BO procedure can be summarized as Algorithm 1.

## 3 Bayesian Kernelized Tensor Factorization for BO

### 3.1 Bayesian Hierarchical Model Specification

Before introducing BKTF, we first construct a $D$-dimensional Cartesian product space corresponding to the search space $\mathcal{X}$. We define it over $D$ sets $\{S_1, \ldots, S_D\}$ and denote as $\prod_{d=1}^{D} S_d$: $S_1 \times \cdots \times S_D = \{(s_1, \ldots, s_D) \mid \forall d \in \{1, \ldots, D\}, s_d \in S_d\}$. For $\forall d \in [1, D]$, the coordinates set $S_d$ is formed by $m_d$ interpolation points that are distributed over a bounded interval $\mathcal{X}_d = [a_d, b_d]$, represented by $\boldsymbol{c}_d = \{c_1^d, \ldots, c_{m_d}^d\}$, i.e., $S_d = \{c_i^d\}_{i=1}^{m_d}$. The size of $S_d$ becomes $|S_d| = m_d$, and the entire space owns $\prod_{d=1}^{D} |S_d|$ samples. Note that $S_d$ could be either uniformly or irregularly distributed.

We randomly sample an initial dataset including $n_0$ input-output data pairs from the pre-defined space, $\mathcal{D}_0 = \{\boldsymbol{x}_i, y_i\}_{i=1}^{n_0}$ where $\{\boldsymbol{x}_i\}_{i=1}^{n_0}$ are located in $\prod_{d=1}^{D} S_d$, and this yields an incomplete $D$-dimensional tensor $\mathcal{Y} \in \mathbb{R}^{|S_1| \times \cdots \times |S_D|}$ with $n_0$ observed points. BKTF approximates the entire data tensor $\mathcal{Y}$ by a kernelized CANDECOMP/PARAFAC (CP) tensor decomposition:

$$\mathcal{Y} = \sum_{r=1}^{R} \lambda_r \cdot \boldsymbol{g}_1^r \circ \boldsymbol{g}_2^r \circ \cdots \circ \boldsymbol{g}_D^r + \boldsymbol{\mathcal{E}}, \tag{6}$$

where $R$ is a pre-specified tensor CP rank, $\boldsymbol{\lambda} = (\lambda_1, \ldots, \lambda_R)^\top$ denote weight coefficients that capture the magnitude/importance of each rank in the factorization, $\boldsymbol{g}_d^r = [g_d^r(s_d) : s_d \in S_d] \in \mathbb{R}^{|S_d|}$ denotes the $r$th latent factor for the $d$th dimension, entries in $\boldsymbol{\mathcal{E}}$ are i.i.d. white noises from $\mathcal{N}(0, \tau^{-1})$. It should be particularly noted that both the coefficients $\{\lambda_r\}_{r=1}^{R}$ and the latent basis functions $\{g_1^r, \ldots, g_D^r\}_{r=1}^{R}$ are random variables. The function approximation for $\boldsymbol{x} = (x_1, \ldots, x_D)^\top$ can be written as:

$$f(\boldsymbol{x}) = \sum_{r=1}^{R} \lambda_r g_1^r(x_1) g_2^r(x_2) \cdots g_D^r(x_D) = \sum_{r=1}^{R} \lambda_r \prod_{d=1}^{D} g_d^r(x_d). \tag{7}$$

For priors, we assume $\lambda_r \sim \mathcal{N}(0, 1)$ for $r = 1, \ldots, R$ and use a GP prior on the latent factors:

$$g_d^r(x_d) \mid l_d^r \sim \mathcal{GP}(0, k_d^r(x_d, x_d'; l_d^r)), \ r = 1, \ldots, R, \ d = 1, \ldots, D, \tag{8}$$

where $k_d^r$ is a valid kernel function. We fix the variances of $k_d^r$ as $\sigma^2 = 1$, and only learn the length-scale hyperparameters $l_d^r$, since the variances of the model can be captured by $\boldsymbol{\lambda}$. One can also exclude $\boldsymbol{\lambda}$ but introduce variance $\sigma^2$ as a kernel hyperparameter on one of the basis functions; however, learning kernel hyperparameter is computationally more expensive than learning $\boldsymbol{\lambda}$. For simplicity, we can also assume the lengthscale parameters to be identical, i.e., $l_d^1 = l_d^2 = \ldots = l_d^R = l_d$, for each dimension $d$. The prior for the corresponding latent factor $\boldsymbol{g}_d^r$ is then a Gaussian distribution: $\boldsymbol{g}_d^r \sim \mathcal{N}(\boldsymbol{0}, \boldsymbol{K}_d^r)$, where $\boldsymbol{K}_d^r$ is the $|S_d| \times |S_d|$ correlation matrix computed from $k_d^r$. We place Gaussian hyperpriors on the log-transformed kernel hyperparameters to ensure positive values, i.e., $\log(l_d^r) \sim \mathcal{N}(\mu_l, \tau_l^{-1})$. For noise precision $\tau$, we assume a conjugate Gamma prior $\tau \sim \text{Gamma}(a_0, b_0)$.

For observations, based on Eq. (7) we assume each $y_i$ in the initial dataset $\mathcal{D}_0$ to be:

$$y_i \mid \{g_d^r(x_d^i)\}, \{\lambda_r\}, \tau \sim \mathcal{N}(f(\boldsymbol{x}_i), \tau^{-1}). \tag{9}$$

### 3.2 BKTF as a Two-layer Deep GP

Here we show the representation of BKTF as a two-layer deep GP. The first layer characterizes the generation of latent functions $\{g_d^r\}_{r=1}^{R}$ for coordinate/dimension $d$ and also the generation of random weights $\{\lambda_r\}_{r=1}^{R}$. For the second layer, if we consider $\{\lambda_r, g_1^r, \ldots, g_D^r\}_{r=1}^{R}$ as parameters and rewrite the functional decomposition in Eq. (7) as a linear function $f(\boldsymbol{x}; \{\xi_r\}) = \sum_{r=1}^{R} \xi_r |\lambda_r| \prod_{d=1}^{D} g_d^r(x_d)$ with $\xi_r \overset{\text{iid}}{\sim} \mathcal{N}(0, 1)$, we can marginalize $\{\xi_r\}$ and obtain a fully symmetric multilinear kernel/covariance function for any two data points $\boldsymbol{x} = (x_1, \ldots, x_D)^\top$ and $\boldsymbol{x}' = (x_1', \ldots, x_D')^\top$:

$$k(\boldsymbol{x}, \boldsymbol{x}'; \{\lambda_r, g_1^r, \ldots, g_D^r\}_{r=1}^{R}) = \sum_{r=1}^{R} \lambda_r^2 \left[ \prod_{d=1}^{D} g_d^r(x_d) g_d^r(x_d') \right]. \tag{10}$$

As can be seen, the second layer has a multilinear product kernel function parameterized by $\{\lambda_r, g_1^r, \ldots, g_D^r\}_{r=1}^R$. There are some properties to highlight: (i) the kernel is **nonstationary** since the value of $g_d^r(\cdot)$ is location-specific, and (ii) the kernel is **nonseparable** when $R > 1$. Therefore, this specification is very different from traditional GP surrogates:

$$\begin{cases} \text{GP with SE-ARD:} & k\left(\boldsymbol{x}, \boldsymbol{x}'\right) = \sigma^2 \prod_{d=1}^D k_d\left(x_d, x_d'\right), \\ & \text{kernel is stationary and separable} \\ \text{additive GP:} & k\left(\boldsymbol{x}, \boldsymbol{x}'\right) = \sum_{d=1}^D k_d^{\text{1st}}\left(x_d, x_d'\right) + \sum_{d=1}^{D-1} \sum_{e=d+1}^D k_d^{\text{2nd}}\left(x_d, x_d'\right) k_e^{\text{2nd}}\left(x_e, x_e'\right), \\ \text{(1st/2nd order)} & \text{kerenl is stationary and nonseparable} \end{cases}$$

where $\sigma^2$ represents the kernel variance, and kernel functions $\left\{k_d(\cdot), k_d^{\text{1st}}(\cdot), k_d^{\text{2nd}}(\cdot), k_e^{\text{2nd}}(\cdot)\right\}$ are stationary with different hyperparameters (e.g., length scale and variance). Compared with GP-based kernel specification, the multilinear kernel in Eq. (10) has a much larger set of hyperparameters and becomes more flexible and adaptive to the data. From a GP perspective, learning the hyperparameter in the kernel function in Eq. (10) will be computationally expensive; however, we can achieve efficient inference of $\{\lambda_r, g_1^r, \ldots, g_D^r\}_{r=1}^R$ under a tensor factorization framework. Based on the derivation in Eq. (10), we can consider BKTF as a "Bayesian" version of the multidimensional Karhunen-Loève (KL) expansion [18], in which the basis functions $\{g_d^r\}$ are random processes (i.e., GPs) and $\{\lambda_r\}$ are random variables. On the other hand, we can interpret BKTF as a new class of stochastic process that is mainly parameterized by rank $R$ and hyperparameters for those basis functions; however, BKTF does not impose any orthogonal constraints on the latent functions.

## 3.3 Model Inference

Unlike GP, BKTF no longer enjoys an analytical posterior distribution. Based on the aforementioned prior and hyperprior settings, we adapt the MCMC updating procedure in [12, 14] to an efficient element-wise Gibbs sampling algorithm for model inference. This allows us to accommodate observations that are not located on the grid space $\prod_{d=1}^D S_d$. The detailed derivation of the sampling algorithm is given in Appendix 7.1.

## 3.4 Prediction and AF Computation

In each step of function evaluation, we run the MCMC sampling process $K$ iterations for model inference, where the first $K_0$ samples are taken as burn-in and the last $K - K_0$ samples are used for posterior approximation. The predictive distribution for any entry $f^*$ in the defined grid space conditioned on the observed dataset $\mathcal{D}_0$ can be obtained by the Monte Carlo approximation $p\left(f^* \mid \mathcal{D}_0, \boldsymbol{\theta}_0\right) \approx \frac{1}{K-K_0} \times \sum_{k=K_0+1}^K p\left(f^* \mid \left(\boldsymbol{g}_d^r\right)^{(k)}, \boldsymbol{\lambda}^{(k)}, \tau^{(k)}\right)$, where $\boldsymbol{\theta}_0 = \{\mu_l, \tau_l, a_0, b_0\}$ is the set of all parameters used in hyperpriors. Although direct analytical predictive distribution does not exist in BKTF, the posterior mean and variance estimated from MCMC samples at each location naturally offer us a Bayesian approach to define the AFs.

BKTF provides a fully Bayesian surrogate model. We define a Bayesian variant of UCB as the AF by adapting the predictive mean and variance (or uncertainty) in ordinary GP-based UCB with the values calculated from MCMC sampling. For every MCMC sample after burn-in, i.e., $k > K_0$, we can estimate a output tensor $\tilde{\boldsymbol{\mathcal{F}}}^{(k)}$ over the entire grid space using the latent factors $\left(\boldsymbol{g}_d^r\right)^{(k)}$ and the weight vector $\boldsymbol{\lambda}^{(k)}$: $\tilde{\boldsymbol{\mathcal{F}}}^{(k)} = \sum_{r=1}^R \lambda_r^{(k)} \left(\boldsymbol{g}_1^r\right)^{(k)} \circ \left(\boldsymbol{g}_2^r\right)^{(k)} \circ \cdots \circ \left(\boldsymbol{g}_D^r\right)^{(k)}$. We can then compute the corresponding mean and variance tensors of the $(K - K_0)$ samples $\{\tilde{\boldsymbol{\mathcal{F}}}^{(k)}\}_{k=K_0+1}^K$, and denote the two tensors by $\boldsymbol{\mathcal{U}}$ and $\boldsymbol{\mathcal{V}}$, respectively. The approximated predictive distribution at each point $\boldsymbol{x}$ in the space becomes $\tilde{f}(\boldsymbol{x}) \sim \mathcal{N}\left(u(\boldsymbol{x}), v(\boldsymbol{x})\right)$. Following the definition of UCB in Eq. (5), we define Bayesian UCB (B-UCB) at location $\boldsymbol{x}$ as $\alpha_{\text{B-UCB}}\left(\boldsymbol{x} \mid \mathcal{D}, \beta, \boldsymbol{g}_d^r, \boldsymbol{\lambda}\right) = u(\boldsymbol{x}) + \beta\sqrt{v(\boldsymbol{x})}$. The next search/query point can be determined via $\boldsymbol{x}_{\text{next}} = \arg\max_{\boldsymbol{x} \in \{\prod_{d=1}^D S_d - \mathcal{D}_{n-1}\}} \alpha_{\text{B-UCB}}(\boldsymbol{x})$.

We summarize the implementation procedure of BKTF for BO in Appendix 7.2 (see Algorithm 2). Given the sequential nature of BO, when a new data point arrives at step $n$, we can start the MCMC with the last iteration of the Markov chains at step $n-1$ to accelerate model convergence. The main computational and storage cost of BKTF is to interpolate and save the tensors $\tilde{\boldsymbol{\mathcal{F}}} \in \mathbb{R}^{|S_1| \times \cdots \times |S_D|}$ over $(K - K_0)$ iterations for Bayesian AF estimation. This could be prohibitive when the MCMC

sample size or the dimensionality of input space is large. To avoid saving the tensors, in practice, we can simply use the maximum values of each entry over the $(K - K_0)$ iterations through iterative pairwise comparison. The number of samples after burn-in then implies the value of $\beta$ in $\alpha_{\text{B-UCB}}$. We adopt this simple AF in our numerical experiments.

## 4  Related Work

The key of BO is to effectively characterize the posterior distribution of the objective function from a limited number of observations. The most relevant work to our study is the *Bayesian Kernelized Factorization* (BKF) framework, which has been mainly used for modeling large-scale and multidimensional spatiotemporal data with UQ. The key idea is to parameterize the multidimensional stochastic processes using a factorization model, in which specific priors are used to encode spatial and temporal dependencies. Signature examples of BKF include spatial dynamic factor model (SDFM) [19], variational Gaussian process factor analysis (VGFA) [20], and Bayesian kernelized matrix/tensor factorization (BKMF/BKTF) [12, 14, 13]. A common solution in these models is to use GP prior to modeling the factor matrices, thus encoding spatial and temporal dependencies. In addition, for multivariate data with more than one attribute, BKTF also introduces a Wishart prior to modeling the factors that encode the dependency among features. A key difference among these methods is how inference is performed. SDFM and BKMF/BKTF are fully Bayesian hierarchical models and they rely on MCMC for model inference, where the factors can be updated via Gibbs sampling with conjugate priors; for learning the posterior distributions of kernel hyperparameters, SDFM uses the Metropolis-Hastings sampling, while BKMF/BKTF uses the more efficient slice sampling. On the other hand, VGFA uses variational inference to learn factor matrices, while kernel hyperparameters are learned through maximum a posteriori (MAP) estimation without UQ. Overall, BKTF has shown superior performance in modeling multidimensional spatiotemporal processes with high-quality UQ for 2D and 3D spaces [14] and conducting tensor regression [13].

The proposed BKTF surrogate models the objective function—as a single realization of a random process—using low-rank tensor factorization with random basis functions. This basis function-based specification is closely related to multidimensional Karhunen-Loève (KL) expansion [18] for stochastic (spatial, temporal, and spatiotemporal) processes. The empirical analysis of KL expansion is also known as proper orthogonal decomposition (POD). With a known kernel/covariance function, truncated KL expansion allows us to approximate the underlying random process using a set of eigenvalues and eigenfunctions derived from the kernel function. Numerical KL expansion is often referred to as the Garlekin method, and in practice the basis functions are often chosen as prespecified and deterministic functions [15, 21], such as Fourier basis, wavelet basis, orthogonal polynomials, B-splines, empirical orthogonal functions, radial basis functions (RBF), and Wendland functions (i.e., compactly supported RBF) (see, e.g., [22], [23], [24], [25]). However, the quality of UQ will be undermined as the randomness is fully attributed to the coefficients $\{\lambda_r\}$; in addition, these methods also require a large number of basis functions to fit complex stochastic processes. Different from methods with fixed/known basis functions, BKTF uses a Bayesian hierarchical modeling framework to better capture the randomness and uncertainty in the data, in which GP priors are used to model the latent factors (i.e., basis functions are also random processes) on different dimensions, and hyperpriors are introduced on the kernel hyperparameters. Therefore, BKTF becomes a fully Bayesian version of multidimensional KL expansion for stochastic processes with unknown covariance from partially observed data, however, without imposing any orthogonal constraint on the basis functions. Following the analysis in section 3.2, BKTF is also a special case of a two-layer deep Gaussian process [26, 10], where the first layer produces latent factors for each dimension, and the second layer holds a multilinear kernel parameterized by all latent factors.

## 5  Experiments

### 5.1  Optimization for Benchmark Test Functions

We test the proposed BKTF model for BO on six benchmark functions that are used for global optimization problems [27], which are summarized in Table 1. Figure 2(a) shows those functions with 2-dimensional inputs together with the 2D Griewank function. All the selected standard functions are multimodal, more detailed descriptions can be found in Appendix 7.4. In fact, we can visually see that the standard Damavandi/Schaffer/Griewank functions in Figure 2(a) indeed have a low-rank

Table 1: Summary of the studied benchmark functions.

| Function | $D$ | Search space | $m_d$ | Characteristics |
|----------|-----|--------------|-------|-----------------|
| Branin | 2 | $[-5, 10] \times [0, 15]$ | 14 | 3 global minima, flat |
| Damavandi | 2 | $[0, 14]^2$ | 71 | multimodal, global minimum located in small area |
| Schaffer | 2 | $[-10, 10]^2$ | 11 | multimodal, global optimum located close to local minima |
| Griewank | 3 | $[-10, 10]^3$ | 11 | multimodal, many widespread and regularly |
| | 4 | $[-10, 10]^4$ | 11 | distributed local optima |
| Hartmann | 6 | $[0, 1]^6$ | 12 | multimodal, multi-input |

structure. For each function, we assume the initial dataset $\mathcal{D}_0$ contains $n_0 = D$ observed data pairs, and we set the total number of query points to $N = 80$ for 4D Griewank and 6D Hartmann function and $N = 50$ for others. We rescale the input search range to $[0, 1]$ for all dimensions and normalize the output data using z-score normalization.

**Model configuration.** When applying BKTF on the continuous test functions, we introduce $m_d$ interpolation points $\boldsymbol{c}_d$ in the $d$th dimension of the input space. The values of $m_d$ used for each benchmark function are predefined and given in Table 1. Setting the resolution grid will require certain prior knowledge (e.g., smoothness of the function); and it also depends on the available computational resources and the number of entries in the tensor which grows exponentially with $m_d$. In practice, we find that setting $m_d = 10 \sim 100$ is sufficient for most problems. We set the CP rank $R = 2$, and for each BO function evaluation run 400 MCMC iterations for model inference where the first 200 iterations are taken as burn-in. We use Matérn 3/2 kernel as the covariance function for all the test functions. Since we build a fully Bayesian model, the hyperparameters of the covariance functions can be updated automatically from the data likelihood and hyperprior.

**Effects of hyperpriors.** Note that in optimization scenarios where the observation data is scarce, the model performance of BKTF highly depends on the hyperprior settings on the kernel length-scales of the latent factors and the model noise precision $\tau$ when proceeding estimation for the unknown points, i.e., $\boldsymbol{\theta}_0 = \{\mu_l, \tau_l, a_0, b_0\}$. A proper hyper-prior becomes rather important. We discuss the effects of $\{\mu_l, \tau_l\}$ in Appendix 7.5.1. We see that for the re-scaled input space, a reasonable setting is to suppose the mean prior of the kernel length-scales is around half of the input domain, i.e., $\mu_l = \log(0.5)$. The hyperprior on $\tau$ impacts the uncertainty of the latent factors, for example, a large model noise assumption allows more variances in the factors. Generally, we select the priors that make the noise variances not quite large, such as the results shown in Figure 4(a) and Figure 5(b) in Appendix. An example of the uncertainty provided by BKTF is explained in Appendix 7.3.

**Baselines.** We compare BKTF with the following BO methods that use GP as the surrogate model. (1) GP $\alpha_{\text{EI}}$: GP as the surrogate model and EI as the AF in continuous space $\prod_{d=1}^{D} \mathcal{X}_d$; (2) GP $\alpha_{\text{UCB}}$: GP as the surrogate model with UCB as the AF with $\beta = 2$, in $\prod_{d=1}^{D} \mathcal{X}_d$; (3) GPgrid $\alpha_{\text{EI}}$: GP as the surrogate model with EI as the AF, in Cartesian grid space $\prod_{d=1}^{D} S_d$; (4) GPgrid $\alpha_{\text{UCB}}$: GP as the surrogate model with UCB as the AF with $\beta = 2$, in $\prod_{d=1}^{D} S_d$. We use the Matérn 3/2 kernel for all GP surrogates. For AF optimization in GP $\alpha_{\text{EI}}$ and GP $\alpha_{\text{UCB}}$, we firstly use the DIRECT algorithm [28] and then apply the Nelder-Mead algorithm [29] to further search if there exist better solutions.

**Results.** To compare optimization performances of different models on the benchmark functions, we consider the absolute error between the global optimum $f^\star$ and the current estimated global optimum $\hat{f}^\star$, i.e., $\left| f^\star - \hat{f}^\star \right|$, w.r.t. the number of function evaluations. We run the optimization 10 times for every test function with a different set of initial observations. The results are summarized in Figure 2(b). We see that for the 2D functions Branin and Schaffer, BKTF clearly finds the global optima much faster than GP surrogate-based baselines. For Damavandi function, where the global minimum ($f(\boldsymbol{x}^\star) = 0$) is located at a small sharp area while the local optimum ($f(\boldsymbol{x}) = 2$) is located at a large smooth area (see Figure 2(a)), GP-based models are trapped around the local optima in most cases, i.e., $\left| f^\star - \hat{f}^\star \right| = 2$, and cannot jump out. On the contrary, BKTF explores the global characteristics of the objective function over the entire search space and reaches the global optimum within 10 iterations of function evaluations. For higher dimensional Griewank and Hartmann functions, BKTF successfully arrives at the global optima under the given observation budgets, while GP-based comparison methods are prone to be stuck around local optima. We illustrate the latent

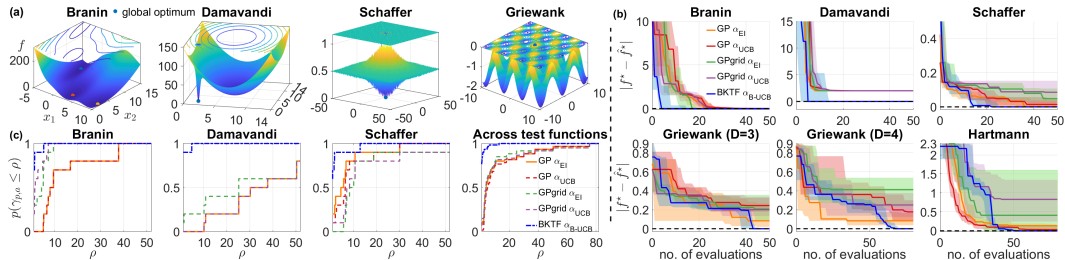

Figure 2: (a) Tested benchmark functions; (b) Optimization results on the six test functions, where medians with 25% and 75% quartiles of 10 runs are compared; (c) Illustration of performance profiles.

Table 2: Results of $\left|f^\star - \hat{f}^\star\right|$ when $n = N$ (mean ± std.) / AUC of PPs on benchmark functions.

| Function ($D$) | GP $\alpha_{EI}$ | GP $\alpha_{UCB}$ | GPgrid $\alpha_{EI}$ | GPgrid $\alpha_{UCB}$ | BKTF $\alpha_{B\text{-}UCB}$ |
|---|---|---|---|---|---|
| Branin (2) | 0.01±0.01/37.7 | 0.01±0.01/37.7 | 0.31±0.62/47.8 | 0.24±0.64/49.2 | **0.00±0.00/50.5** |
| Damavandi (2) | 2.00±0.00/17.6 | 2.00±0.00/17.6 | 1.60±0.80/24.2 | 2.00±0.00/17.6 | **0.00±0.00/50.6** |
| Schaffer (2) | 0.02±0.02/44.9 | 0.02±0.02/43.1 | 0.10±0.15/38.3 | 0.09±0.07/38.0 | **0.00±0.00/49.6** |
| Griewank (3) | 0.14±0.14/48.9 | 0.25±0.10/47.7 | 0.23±0.13/47.7 | 0.22±0.12/47.7 | **0.00±0.00/50.8** |
| Griewank (4) | 0.10±0.07/79.5 | 0.19±0.12/77.8 | 0.38±0.19/77.8 | 0.27±0.17/77.8 | **0.00±0.00/80.5** |
| Hartmann (6) | 0.12±0.07/78.0 | 0.07±0.07/78.0 | 0.70±0.70/79.1 | 0.79±0.61/78.9 | **0.00±0.00/80.7** |
| Overall | -/70.3 | -/69.53 | -/71.3 | -/70.4 | **-/80.5** |

Best results are highlighted in bold fonts.

factors of BKTF for 3D Griewank function in Appendix 7.5.3, which shows the periodic (global) patterns automatically learned from the observations. We compare the absolute error between $f^\star$ and the final estimated $\hat{f}^\star$ in Table 2. The enumeration-based GP surrogates, i.e., GPgrid $\alpha_{EI}$ and GPgrid $\alpha_{UCB}$, perform a little better than direct GP-based search, i.e., GP $\alpha_{EI}$ and GP $\alpha_{UCB}$ on Damavandi function, but worse on others. This means that the discretization, to some extent, offers possibilities for searching all the alternative points in the space, since in each function evaluation, every sample in the space is equally compared solely based on the predictive distribution. Overall, BKTF reaches the global optimum for every test function and shows superior performance for complex objective functions with a faster convergence rate. To intuitively compare the overall performances of different models across multiple experiments/functions, we further estimate performance profiles (PPs) [30] (see Appendix 7.5.2), and compute the area under the curve (AUC) for quantitative analyses (see Figure 2(c) and Table 2). Clearly, BKTF obtains the best performance across all functions.

## 5.2 Hyperparameter Tuning for Machine Learning

In this section, we evaluate the performance of BKTF for automatic machine-learning tasks. Specifically, we compare different models to optimize the hyperparameters of two machine learning (ML) algorithms—random forest (RF) and neural network (NN)—on classification for the MNIST database of handwritten digits[1] and housing price regression for the Boston housing dataset[2]. The details of the hyperparameters that need to learn are given in Appendix 7.6. We assume the number of data points in the initial dataset $\mathcal{D}_0$ equals the dimension of hyperparameters need to tune, i.e., $n_0 = 4$ and $n_0 = 3$ for RF and NN, respectively. The total budget is $N = 50$. We implement the RF algorithms using scikit-learn package and construct NN models through Keras with 2 hidden layers. All other model hyperparameters are set as the default values.

**Model configuration.** We treat all the discrete hyperparameters as samples from a continuous space and then generate the corresponding Cartesian product space $\prod_{d=1}^{D} S_d$. One can interpret the candidate values for each hyperparameter as the interpolation points in the corresponding input dimension. According to Appendix 7.6, the size of the spanned space $\prod S_d$ is $91 \times 46 \times 64 \times 10$ and $91 \times 46 \times 13 \times 10$ for RF classifier and RF regressor, respectively; for the two NN algorithms, the size of parameter space is $91 \times 49 \times 31$. Similar to the settings on standard test functions, we set tensor rank $R = 2$, set $K = 400$ and $K_0 = 200$ for MCMC inference, and use the Matérn 3/2 kernel.

---

[1] http://yann.lecun.com/exdb/mnist/

[2] https://www.cs.toronto.edu/~delve/data/boston/bostonDetail.html

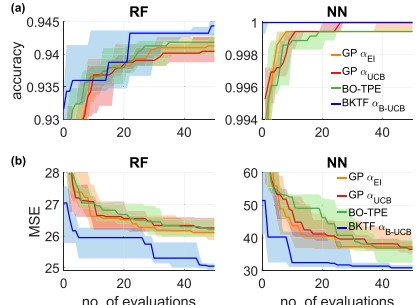

| Final accuracy for (a) and MSE for (b). | | |
|---|---|---|
| Model | RF | NN |
| (a) GP $\alpha_{EI}$ | 94.12±0.16 | 99.96±0.03 |
| GP $\alpha_{UCB}$ | 94.07±0.19 | 99.99±0.02 |
| BO-TPE | 94.14±0.20 | 99.96±0.02 |
| BKTF $\alpha_{B-UCB}$ | **94.44**±0.15 | **100.00**±0.00 |
| (b) GP $\alpha_{EI}$ | 26.19±0.45 | 38.46±3.31 |
| GP $\alpha_{UCB}$ | 26.29±0.35 | 36.78±1.91 |
| BO-TPE | 26.27±0.31 | 36.40±4.72 |
| BKTF $\alpha_{B-UCB}$ | **25.03**±0.18 | **30.84**±1.13 |

The values are presented as mean±std.
Best results are highlighted in bold fonts.

Figure 3 & Table 3: Results of hyperparameter tuning for automated ML: (a) MNIST classification; (b) Boston housing regression. The figure compares medians with 25% and 75% quartiles of 10 runs.

**Baselines.** Other than the GP surrogate-based GP $\alpha_{EI}$ and GP $\alpha_{UCB}$, we also compare with Tree-structured Parzen Estimator (BO-TPE) [31], which is a widely applied BO approach for hyperparameter tuning. We exclude grid-based GP models as sampling the entire grid becomes infeasible.

**Results.** We compare the accuracy for MNIST classification and MSE (mean squared error) for Boston housing regression both in terms of the number of function evaluations and still run the optimization processes ten times with different initial datasets $\mathcal{D}_0$. The results obtained by different BO models are given in Figure 3, and the final classification accuracy and regression MSE are compared in Table 3. For BKTF, we see from Figure 3 that the width between the two quartiles of the accuracy and error decreases as more iterations are evaluated, and the median curves present superior convergence rates compared to baselines. For example, BKTF finds the hyperparameters of NN that achieve 100% classification accuracy on MNIST using less than four function evaluations in all ten runs. Table 3 also shows that the proposed BKTF surrogate achieves the best final mean accuracy and regression error with small standard deviations. All these demonstrate the advantage of BKTF as a surrogate for black-box function optimization.

## 6 Conclusion and Discussions

In this paper, we propose to use Bayesian Kernelized Tensor Factorization (BKTF) as a new surrogate model for Bayesian optimization. Compared with traditional GP surrogates, the BKTF surrogate is more flexible and adaptive to data thanks to the Bayesian hierarchical specification, which provides high-quality UQ for BO tasks. The tensor factorization model behind BKTF offers an effective solution to capture global/long-range correlations and cross-dimension correlations. Therefore, it shows superior performance in characterizing complex multidimensional stochastic processes that are nonstationary, nonseparable, and multimodal. The inference of BKTF is achieved through MCMC, which provides a natural solution for acquisition. Experiments on both test function optimization and ML hyperparameter tuning confirm the superiority of BKTF as a surrogate for BO. A limitation of BKTF is that we restrict BO to Cartesian grid space to leverage tensor factorization; however, we believe designing a compatible grid space based on prior knowledge is not a challenging task.

There are several directions to be explored for future research. A key computational issue of BKTF is that we need to reconstruct the posterior distribution for the whole tensor to obtain the AF. This could be problematic for high-dimensional problems due to the curse of dimensionality. It would be interesting to see whether we can achieve efficient acquisition directly using the basis functions and corresponding weights without constructing the tensors explicitly. In terms of rank determination, we can introduce the multiplicative gamma process prior to learn the rank; this will create a Bayesian nonparametric model that can automatically adapt to the data. In terms of surrogate modeling, we can further integrate a local (short-scale) GP component to construct a more precise surrogate model, as presented in [14]. The combined framework would be more expensive in computation, but we expect the combination to provide better UQ performance. In terms of parameterization, we also expect that introducing orthogonality prior to the latent factors (basis functions) will improve the inference. This can be potentially achieved through more advanced prior specifications such as the Matrix angular central Gaussian [32]. In addition, for the tensor factorization framework, it is straightforward to adapt the model to handle categorical variables as input and multivariate output by placing a Wishart prior to the latent factors for the categorical/output dimension.

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
