# OpenReview forum: "Bayesian Kernelized Tensor Factorization as Surrogate for Bayesian Optimization"
_NeurIPS.cc/2023/Conference — Submitted to NeurIPS 2023_

### Official Review · Reviewer_LjoC · 2023-06-20

**Soundness:** 1 poor
**Presentation:** 2 fair
**Contribution:** 3 good
**Rating:** 4
**Confidence:** 4

**Summary:**

This proposes a novel surrogate for Bayesian optimization (BO), kernelized tensor factorization (BKTF). The authors claim that BKTF is able to model more complex functions (nonstationary, nonseparable) compared to additive and product kernel Gaussian processes. For inference, they leverage Gibbs sampling to do full Bayesian inference. They compare against BO with the regular Gaussian process surrogate and tree-structured Parzen estimators.

**Strengths:**

* The papers proposes a novel surrogate model, BKTF, for BO. I believe this is new, and as long as the authors can demonstrate the utility of BKTF, this will be a valuable contribution to the BO community.

**Weaknesses:**

* The proposed strategy uses Gibbs sampling, which is well known scale poorly with the number of parameters and correlations. Thus, I am concerned that this method's performance will fair poorly at even moderately higher dimensions and number of observations than those considered in this paper.
* On a similar note, unless I'm not mistaken, the method requires to infer the latent functions (or bases) $g_d^D$. This means inference needs to be performed at every BO steps. This contrast to GPs, where, even if one decides to do fully Bayesian inference, he does not to run MCMC at every step. Thus, the method comes with a reduction in flexibility. If the authors believe that their method can work with less expensive inference strategies, say, VI or MAP, then this should be demonstrated and evaluated.
* The paper claims that the experiments are "extensive" (line 73), but unfortunately, I find that the experiments conducted in this paper cannot be considered to be extensive in today's standards. See for example [1,2], which I would consider extensive. Furthermore, at this small scale / low budget applications, noise can very easily swamp the effects. Therefore, I would expect a lot more runs. Moreover, the hyperparameter tuning experiments in Section 5.2 are not reflective of real-world use cases. So these are gain inadequate to evaluate the real world performance of BKTF.
* Furthermore, the baselines are not enough. The research space for alternatives to BO surrogates has certainly been active, but here only the tree-structured Parzen estimator is considered. In fact, the paper mentions that BKTF here corresponds to a two-layer deep GP. Then, they should compare against deep GPs for an apple-to-apple comparison. The computational costs/scalability of DGPs would probably be comparable so this would be a more appropriate comparison.

**Questions:**

### Additional Major Comments
* The paper repeatedly uses the term "separable" to characterize the SE-ARD kernel. I have never seen "separable" used in the context other than "additively separable." Multiplicative kernels model the correlations between dimensions, thus I'm not sure what BKTF is doing more than SE-ARD.
* Line 49-50: The paper claims that deep GPs require a large number of samples. I presume this statement is relative to BKTF, but I do not find evidence in this paper that BKTF require less samples.
* Line 251: What does "low-rank" mean in this context?
* The paper does not cite the original source when referring to existing methods. For example:
  * Line 56: CANDECOMP, PARAFAC
  * Line 69: Slice sampling
  * Line 117: I think the GP-UCB paper [3] should be cited here.
* Line 115: $\beta$ in GP-UCB is not necessarily a tunable parameter in the sense that, the optimal configuration, under assumptions, is very specific. See [3].
* The papers applied UCB acquisition functions to their method, but UCB is known to be conservative, and not that competitive among acquisition functions. Thus, I recommend using other acquisition functions.
* Section 4: Considering that this paper proposes for an alternative surrogate, the related work section should put the proposed method in context of alternative surrogates. Unfortunately, the current form mostly discusses kernel factorization, which I think is less useful for the BO community. After all, the papers on BKTF referenced herein could be referred to obtain context.

### Minor Comments
* Above Line 158: kerenl $\to$ kernel.
* Between Line 87 and 88: Normally, we denote that the expectation of the noisy version of $f$, for example, $\hat{f} = f + \epsilon$, is minimized instead of saying that we optimize $f$ directly.


### References
I am not affiliated with any of the papers and authors mentioned in this review.
1. [1] Bodin, Erik, et al. "Modulating surrogates for Bayesian optimization." International Conference on Machine Learning. PMLR, 2020.
2. [2] Malkomes, Gustavo, and Roman Garnett. "Automating Bayesian optimization with Bayesian optimization." Advances in Neural Information Processing Systems 31 (2018).
3. [3] Srinivas, Niranjan, et al. "Information-theoretic regret bounds for gaussian process optimization in the bandit setting." IEEE transactions on information theory 58.5 (2012): 3250-3265.


**Limitations:**

Yes, in Section 6. However, I think the limitations I've discussed above could also be included.

---

> ### Author Rebuttal · Authors · 2023-08-10
>
> We appreciate your time and thank you for your review and your acknowledgment of the contribution of this model. Through the review, however, we believe there is some misunderstanding and misinterpretation of the paper. In the response below, we will try to explain and clarify these points as much as we can.
>
> $\textbf{for Weaknesses:}$
> 1. We have discussed the computational cost of BKTF with respect to model inference and AF computation in response to all. We have to say that Gibbs sampling is not the reason for the scalability issue; instead, we consider MCMC the best way to achieve uncertainty quantification for complex models. The main computation bottleneck is the grid enumeration-based AF computation; we have considered randomly selecting candidate points for AF as an alternative solution for higher dimensional problems and tested it on benchmark functions. We have also tested on a higher dimensional 10D Griewank function in the experiments, see Table A and Figure A in PDF. We think the newly added experiments could explain the doubts about the scalability of data dimensions and observation points. We will also add the average running time per function evaluation later. As can be seen, BKTF consistently obtains the best performance on all tested functions, which demonstrates its efficiency and effectiveness to solve BO problems. About building a fully Bayesian model: we want to highlight that it is not the problem but is one key contribution of this work. Fully Bayesian makes BKTF generate high-quality uncertainty totally from observations, which provides candidates that leverage global consistency from the data and make it possible to achieve efficient global search with a much less budget.
> 2. $\boldsymbol{g}_d^r$ do need to be sampled, but they can be updated only using the observation points, see the model inference in Appendix. However, we argue that this does not affect the computational cost of model inference, which is still $\mathcal{O}\left(n^3\right)$ if utilizing point-wise sampling. For "even if one decides to do fully Bayesian... does not run MCMC at every step": We disagree. We have to say once mentioned fully Bayesian, with the inclusion of a new data point, the model parameters and hyperparameters need to be updated. "with a reduction in flexibility": We do not see why fully Bayesian inference reduces flexibility. As for VI/MAP, again the inference is not the main cost, see our discussion in response to all. Even though these algorithms have the potential of reducing computational cost, however,  at the cost of the quality of uncertainty quantification, which is critical to the efficiency of BO. This has also been discussed in related work that develop fully Bayesian AF/GP for BO.
> 3. We have conducted more experiments, including higher dimensional functions and more baselines. We conduct 7 benchmark functions, 4 ML hyperparameter tuning tasks, and 1 synthetic process (in Introduction). For the "noise", we believe whether the true function has noises or not, this will not affect the result. Since in the estimation, we always assume noisy data no matter how the true data is generated. The advantage of BKTF is the same. "experiments in Section 5.2...not reflective...": We are extending the experiments for this section in terms of adding more baselines and catigorical input variables.The considered settings for the ML hyperparameters are given in Table 4 in Appendix; we think these tasks are similar to the experiments in the second reference you gave and the similar settings have also been tested in related work.
> 4. We added more baselines, including additive GP, Bayesian neural networks, and non-GP approaches. But we did not compare with deep GP, since it does not have analytical mean and uncertainty and cannot provide closed-form AF equations. One advantage of BKTF is that it is flexible and the model can be efficiently sampled leveraging the tensor factorization structure, which is however difficult to be implemented for deep GP.
>
> $\textbf{for Questions: Major comments:}$
> 1. We have discussed the nonstationary and nonseparable processes in the response to all. A kernel/covariance function is separable if it can be decomposed into a product of functions along each dimension. SE-ARD is a stationary and separable kernel. The kernel representation of BKTF shows that the covariance has a sum structure when $R\ge 2$ (thus nonseparable) and location-specific due to the latent basis (thus nonstationary).
> 2. BKTF indeed requires fewer samples to achieve better results compared with other surrogates (although we do not have deep GP), as demonstrated in almost all experiments. This is the key advantage of the model.
> 3. It means the true functions are not full-rank and have global patterns.
> 4. Thank you, we will cite the corresponding references in the revised version.
> 5. We agree that $\beta$ is not a tunable parameter, we actually did not tune it.
> 6. Bayesian UCB is a natural solution given the MCMC samples we obtain through model inference. It should be noted that other AF (e.g., EI) are not considered because we do not have analytical uncertainty.
> 7. We agree that we should cite more related work about BO and have revised the paper, thank you; but we think the discussion about kernel representation is also critical and is a problem that might have been ignored in the BO community, which can be a simple but efficient strategy to improve the performance.
>
> $\textbf{Minor Comments:}$ Thanks for the detailed comments.
> 8. the typo has been corrected, we have also re-checked the whole paper.
> 9. in the paper, we denote the observation data $y$ as the noisy version of $f$.

---

> > ### Comment · Reviewer_LjoC · 2023-08-12
> > **Response**
> >
> > I sincerely thank the authors for their response. I also appreciate the additional experiments. However, I find that there are some disagreements, especially about the utility of the proposed method, which I wish to highlight here.
> >
> > >  About building a fully Bayesian model: we want to highlight that it is not the problem but is one key contribution of this work. Fully Bayesian makes BKTF generate high-quality uncertainty totally from observations, which provides candidates that leverage global consistency from the data and make it possible to achieve efficient global search with a much less budget.
> > >
> > > Even though these algorithms have the potential of reducing computational cost, however, at the cost of the quality of uncertainty quantification, which is critical to the efficiency of BO. This has also been discussed in related work that develop fully Bayesian AF/GP for BO.
> > >
> > > We do not see why fully Bayesian inference reduces flexibility.
> >
> > I do not think that being fully Bayesian is bad. However, there is no clear conclusion about the benefit of being fully Bayesian in the BO setting. See the works of De Ath et al. (2021), Berkenkamp et al. (2019). And trust me; I am as Bayesian as you can get. But specifically for BO, I think it is important to acknowledge that the benefit of being fully Bayesian is not yet entirely clear, and therefore provide alternative scalable strategies such as MAP. Furthermore, variational inference (VI), and Laplace approximation are fully Bayesian, and potentially much more efficient than Gibbs.
> >
> > > We will also add the average running time per function evaluation later.
> > >
> > > As for VI/MAP, again the inference is not the main cost, see our discussion in response to all.
> >
> > I think this would be essential if the authors wish to resolve any doubts about the scalability of the method. I certainly believe the authors. However, what we need here is scientific evidence. In particular, the computational cost needs to be compared against *non-fully-Bayesian* GP methods (MAP-II), which are widely used.
> >
> > Furthermore, in terms of computation, I checked the code, and it seems that the authors are using random-walk Metropolis-Hastings (RWMH) for the GP baselines. This is *very* concerning. Excuse me if I missed this detail somewhere, but the paper does not seem to report how well MCMC is converging, specifically, $\widehat{R}$ metrics, effective sample sizes, and average acceptance rates. This is important because Gibbs sampling is insensitive to hyperparameters while RWMH is very sensitive. (I would like to note that in most BO papers NUTS and slice sampling are mostly used for the tuning issues.)
> >
> > > We added more baselines, including additive GP, Bayesian neural networks, and non-GP approaches. But we did not compare with deep GP, since it does not have analytical mean and uncertainty and cannot provide closed-form AF equations. One advantage of BKTF is that it is flexible and the model can be efficiently sampled leveraging the tensor factorization structure, which is however difficult to be implemented for deep GP.
> >
> > I appreciated the additional experiments, but I'm not convinced why deep GPs cannot be used. Deep GPs have been used for BO before and can be used with Monte Carlo acquisition functions. I think comparing with deep GPs is critical here, because it is the closest baseline in literature to the proposed method.
> >
> > > The definition of non-stationary and non-separable processes.
> >
> > Can the authors specifically clarify whether this term has been used in the literature before? I am confused that the authors use the word independent here; the kernel is multiplicative across the dimensions how can be this independent?
> >
> > ### References
> > George De Ath, Richard M. Everson, and Jonathan E. Fieldsend. 2021. How Bayesian should Bayesian optimisation be? In Proceedings of the Genetic and Evolutionary Computation Conference Companion (GECCO '21). Association for Computing Machinery, New York, NY, USA, 1860–1869. https://doi.org/10.1145/3449726.3463164
> >
> > Berkenkamp, Felix, Angela P. Schoellig, and Andreas Krause. "No-Regret Bayesian Optimization with Unknown Hyperparameters." Journal of Machine Learning Research 20 (2019): 1-24.
> >
> > Balandat, Maximilian, Brian Karrer, Daniel Jiang, Samuel Daulton, Ben Letham, Andrew G. Wilson, and Eytan Bakshy. "BoTorch: A framework for efficient Monte-Carlo Bayesian optimization." Advances in neural information processing systems 33 (2020): 21524-21538.

---

> > > ### Author Response · Authors · 2023-08-15
> > >
> > > Thank you for your reply. We discussed and replied below, hope can address your concerns.
> > > 1. It seems that a major concern is why we stick to the "fully Bayesian" method in BKTF. The main reason, which we believe has been highlighted in our paper, is that BKTF does not have analytical posterior distribution to support uncertainty quantification (UQ). This is the key difference from GP, where analytical posterior can be used for BO tasks.
> > >
> > > The main reason why not using MAP is that MAP only provides a point estimate without full UQ. Thus estimating BKTF with MAP cannot support BO. Again, given the simplicity of tensor decomposition model, most variables enjoy analytical posterior and it becomes natural to use MCMC.
> > >
> > > For VI, we agree that it is possible to use VI and it has been used in the GPFA paper([R1] J. Luttinen et al., “Variational Gaussian-process factor analysis for modeling spatio-temporal data,” NeurIPS, 2009) to learn latent factors. However, the estimation for kernel hyperparameter is still MAP without UQ. Two reasons we prefer to not use VI for BKTF. Firstly, VI/Laplace involves approximations for Bayesian inference (which we do not think can be called fully Bayesian; fully Bayesian indicates directly updating from the posterior). Generally, simplified assumptions are made for the posterior distributions for model parameters, such as the independent assumptions between variables in posteriors in VI, which may overlook some posterior correlations, obtain inaccurate posteriors, and largely impact the quality of UQ particularly when the data size is small (see [R2] A. Sauer et al., “Non-stationary Gaussian Process surrogates,” arXiv preprint, 2023). As high-quality uncertainty is the most crucial part of BO, the poor uncertainty of VI will undermine the advantage of BKTF and affect the global search efficiency for BO. Secondly, As we know, VI is mainly used for a large number of observations. We have explained the computational cost for this work: about model inference, the cost is at least better than GP inference, and is not the main cost for BO tasks that only has hundreds of observations. Specifically, here observations are less than 100 for most of the tested functions. The average running time of BKTF per evaluation for 2D Branin is 0.35s; even for 10D Griewank, the time is 2.33s, which we believe is acceptable and really not a problem here. In such cases, we do not see the benefit of having the efficiency of VI but losing the high-quality UQ of MCMC. Overall, from our results and the conclusions in related studies, MCMC indeed provides better performance in BO; MCMC is not a problem for BKTF and we develop an efficient sampling algorithm; MCMC is important for BKTF to achieve the current results. Therefore we suggest using MCMC in the proposed framework. We will definitely consider VI in future research if it is really needed (e.g. when having a large data set), we will add the discussion about VI in revised paper.
> > >
> > > 2. "...RWMH for...'': We'd like to clarify that BKTF is implemented exactly as described in the paper (and appendix): The latent factors and weights are updated with Gibbs sampling, and the GP kernel hyperparameters are updated with slice sampling. We did not use RWMH in the model.
> > >
> > > 3. "...deep GPs...": We have tried to use deep GP here; it does not have analytical uncertainty and needs approximation methods for AF. We did consider Bayesian deep GP with MCMC AFs (as you mentioned). The main problem is that the model parameters (multivariate variables) in latent layers do not have analytical posteriors and a complicated sampling algorithm is required. In a recent and relevant work [R3] A. Sauer et al., “Active learning for deep Gaussian Process surrogates,” Technometrics, 2023, ESS is used to address this problem, which still needs more than thousands of MCMC samples for inference per evaluation. In contrast, BKTF has analytical posteriors for the latent factors ${g}_d^r$ (a closed-form Gaussian), which can be updated easily and efficiently. Given the time limit, we could not adapt [R3] for BO tasks and thus did not complete the experiments. However, we believe that BKTF can be seen as a more elegant framework that achieves similar model flexibility as a two-layer deep GP but with much better sampling/inference efficiency (less time cost). We will add a more detailed discussion about the connection with deep GP in the revised paper.
> > >
> > > 4. "non-stationary and non-separable...?" Yes, certainly. Stationarity and separability are important concepts when defining covariance/kernel functions. We refer the reviewer to [R4] M. G. Genton et al., “Cross-covariance functions for multivariate geostatistics,” Statistical Science, 2015 for a full review of related definitions. For a more recent reference in machine learning, we refer the reviewer to [R5] K. Wang et al., “Nonseparable non-stationary random fields,” ICML, 2020. We will better introduce related definitions if we can revise the paper.

---

> > > > ### Comment · Reviewer_LjoC · 2023-08-16
> > > > **Further Comment**
> > > >
> > > > Thank you for your response.
> > > >
> > > > > "...RWMH for...'': We'd like to clarify that BKTF is implemented exactly as described in the paper (and appendix): The latent factors and weights are updated with Gibbs sampling, and the GP kernel hyperparameters are updated with slice sampling. We did not use RWMH in the model.
> > > >
> > > > Yes, I am aware that the authors have used Gibbs sampling for BKTF. I am referring to the GP *baselines*. What did you use for hyperparameter tuning? It doesn't seem to be stated in Section 5.1, Paragraph Baselines. I assumed the authors used fully Bayesian inference here, too, using the file `hyperMatern.m`, which looks like RWMH. If the authors did not use this, please state what you used for tuning the GP baselines.
> > > >
> > > > > "non-stationary and non-separable...?" Yes, certainly. Stationarity and separability are important concepts when defining covariance/kernel functions. We refer the reviewer to [R4] M. G. Genton et al., “Cross-covariance functions for multivariate geostatistics,” Statistical Science, 2015 for a full review of related definitions. For a more recent reference in machine learning, we refer the reviewer to [R5] K. Wang et al., “Nonseparable non-stationary random fields,” ICML, 2020. We will better introduce related definitions if we can revise the paper.
> > > >
> > > > Thank you! Please cite these in the paper. Because in the scalable GP/BO literature, the term "non-separable" is not standard! Instead, the term "product kernel" has been used before (Garnder, 2018). Also, to my knowledge, the widely used Matern 5/2 kernel is not separable (product kernel).
> > > >
> > > > > The main reason why not using MAP is that MAP only provides a point estimate without full UQ. Thus estimating BKTF with MAP cannot support BO. Again, given the simplicity of tensor decomposition model, most variables enjoy analytical posterior and it becomes natural to use MCMC.
> > > >
> > > > Yes! This is exactly what I meant by reduced flexibility in my original review! Unlike GPs, where empirical Bayes can be done efficiently, you don't have this option with BKTF.
> > > >
> > > > ### References
> > > > Gardner, Jacob, Geoff Pleiss, Ruihan Wu, Kilian Weinberger, and Andrew Wilson. "Product kernel interpolation for scalable Gaussian processes." In International Conference on Artificial Intelligence and Statistics, pp. 1407-1416. PMLR, 2018.

---

> > > > > ### Author Response · Authors · 2023-08-17
> > > > >
> > > > > Thank you for your comment.
> > > > >
> > > > > 1. >I am referring to the GP baselines. What did you use for hyperparameter tuning? It doesn't seem to be stated in Section 5.1, Paragraph Baselines. I assumed the authors used fully Bayesian inference here, too, using the file hyperMatern.m, which looks like RWMH. If the authors did not use this, please state what you used for tuning the GP baselines.
> > > > >
> > > > > For the GP-based baselines, we follow the common approach to learn kernel hyperparameters of GP by optimizing log marginal likelihood. As GP already provides analytical predictive distribution (with mean and variance/uncertainty) by default, we did not use fully Bayesian for the GP-based baselines. The "$\textit{hyperMatern.m}$" function is slice sampling used for learning the hyperparameters for BKTF. We did not give a full description of the code in the submitted paper given the page limit. We will add a related statement in the paper and provide a detailed description when releasing the code publicly.
> > > > >
> > > > > 2. We will cite the papers/references and revise the paper correspondingly. Hope our answers have addressed your other concerns and you can consider raising your score for this work, any other questions or comments are welcomed. Thank you again.

---

> > > > > > ### Comment · Reviewer_LjoC · 2023-08-17
> > > > > > **Response**
> > > > > >
> > > > > > I sincerely thank the authors for the many clarifications and continuous engagement. In light of the additional larger-scale experiments, I am willing to raise my score. However, I am still leaning towards rejection because I feel that the baselines are not in the highest standards for this line of work. Not only the discussions about DGPs, but given that the method employs fully Bayesian inference, the GP baselines should also have included cases where the hyperparameters are dealt in a fully Bayesian way. Nevertheless, I encourage the authors to include the limitations that I have highlighted.

---

### Official Review · Reviewer_1Atp · 2023-06-25

**Soundness:** 3 good
**Presentation:** 2 fair
**Contribution:** 3 good
**Rating:** 5
**Confidence:** 3

**Summary:**

Bayesian optimisation most commonly uses Gaussian Processes with the Squared exponential or Matern kernel as the surrogate model. The authors propose a new type of surrogate model, "Bayesian Kernelized Tensor Factorization" which introduces some advantages and disadvantages over Gaussian Processes. There seems to be prior work investigating these models for surrogate modding in general, and this paper is a followup applying these models within Bayesian optimisation in particular.

The papers introduces the model which models the data $\{x_i, y_i}$ from a black box function $y=f(x)$ as a sum of functions where each function is a product of 1 dimensional GPs, e.g. in 2D, leteach $g()$ be a 1D GP then
$$
\hat{f}(x_1, x_2) = \sum_i g^i_1(x_1)g^i_2(x_2)
$$
which is a continuous analogue of how a matrix can be represented by it's SVD or eigen decomposition. This concept generalizes for multiple input dimensions (e.g. $g^i_1(x_1)g^i_2(x_2)g^i_3(x_3)g^i_4(x_4).....$) and the authors discretize the search space into a grid hence the implementation uses tensors.
 and it positive properties,
- to be able to model function with separability (where variables do not interact like in additive kernels) and
- non-stationarity.

In my interpretation, the thesis of the paper is that these properties are significant disdvantes and using a model that has these properties enables a performance improvement.

The disadvantage of the proposed model is that inference is no longer in closed form (a product of Gaussian random variables is not another Gaussian) hence an MCMC method is proposed to sample function values at points across the input space. While one could use a random discretization, (e.g. a latin hypercube, or a cluster around the current best point)  given the product structure of the surrogate model, there appear to be implementation benefits using tensor and matrix Kronecker products if the discretization is a fixed grid, discretize each dimension and build a a Cartesian product of each dimension to have a full grid.

A range of synthetic and hyperparameter tuning benchmarks show the new model performing favourably with standard GP using SE-ARD kernel.

**Strengths:**

- in theory, I really like the idea of the model, in particular, that any matrix can be decomposed by SVD, hints that any function can be decomposed into a sum and product over functions of each dimension, i.e. the proposed model is, in theory, a universal approximator? Although intuitively, both BKTF and SE-ARD can model any smooth non-stationary surface, discontinuities and kinks are not modellable.

- the inclusion of grid based GP methods is nice to see, and shows how much the grid decays performance compared to using the full unrestricted continuous space

**Weaknesses:**

# Technical
- the proposed Cartesian discretization, $S_D$, scales exponentially with input dimension, and presumably contains _a lot_ of useless points in empty parts of the search space would a random discretization (LHC or Gaussian around current best $x$) be so much worse? Given a random set of points $X_D$, it is trivial to compute a the joint prior density $P[f(X_D)]$ density and the likelihood is just Gaussian $P[y_i|f(x_i)]$, sampling function values can be done with any off-the-shelf MCMC method.

- I believe at least an additive GP should be a baseline. If non-stationary and non-separability are the main advantages of the BKTF model, presumably an additive GP with 2 kernels per dimension (matching CP rank=2 for BKTF) is an obvious baseline that has separability, such a baseline is exactly equation (7) but with sum-sum instead of sum-product. From this perspective, BKTF is simply an additive GP (that can only model separable variables) with a product over dimensions instead of a sum and this one change introduces a lot of engineering overhead (MCMC inference vs closed form inference) but also introduces more modelling power (separability can be modelled), given a high enough CP rank (CP rank =1 is just a product of 1D funs and is not separable).


- the related work consists of two paragraphs, the first is discusses prior work on  BKTF (and feels a bit repetitive), the second focuses on stochastic process models. I feel the novelty of this paper is in using another surrogate model inside BO methods, and given the large body of BO work, there have been many works acknowledginbg the limitations of SE-ARD and proposing alternative models that are not cited or empirically compared to
  - [Bayesian Neural networks](https://scholar.google.com/scholar?hl=en&as_sdt=0%2C5&q=bayesian+optimization+neural+networks&btnG=):
    - Bayesian optimization with robust Bayesian neural networks, NeurIPS 2016
    - Scalable bayesian optimization using deep neural networks, ICML 2015
    - Multi-fidelity Bayesian optimization via deep neural networks: NeurIPS 2020
  - [Deep GPs](https://scholar.google.com/scholar?hl=en&as_sdt=0%2C5&q=bayesian+optimization+deep+gaussian+&btnG=)
    - Bayesian optimization using deep Gaussian processes, Arxiv

# Presentation

(in my personal subjective view) some changes to presentation would have made the paper far more accessible to me
- can "CANDECOMP/PARAFAC" be simply described as a tensor generalization of SVD to make it easier for readers?
- L119: "we construct a D-dimensinoal Cartesian product Space", can we just say "grid" like the authors do for the rest of the paper?
- L81: kronecker product is introduced and never used again in the main paper
- Section 3.1 would be much easier for me to understand if Eq (7) and (8) are introduced first, then next Section 3.3 (model inference) describes the grid and Equation (6) and MCMC details and the justification for the grid.
- Section 3.2 is nice to mention but for me distracts from the main paper hence would be much better suited to the appendix.
- L192: given a mean and uncertainty, this seems to be standard UCB, why is "Bayesian-UCB" defined?

**Questions:**

- the main body of the paper as it is makes the grid feel unnecessary in my view, the paper lacks a justification. Is the grid discretization _really_ required? L172 acknowledges a grid is not required, it is not _required_ for MCMC, the model structure, or for BO. It seems the grid is an implementation choice that does make nice use of the kronecker structure in the model but also introduces a exponential scaling limitations (12**6  = 3m points in the Hartmann experiment). Would the authors mind adding a "memory used" or "time consumed" column to Tables 2 and 3 top show practical implications of the discretization? Or add baseline with a randomized discretizations instead?

- can the authors include 1D additive GPs as a baseline, even better adding two kernels (with independent hyperparameters) per dimension would be a very close model the BKTF with CP rank=2

- at least one Bayesian neural network baseline model would compare this surrogate model with other well studied non-GP surrogate models, e.g. https://github.com/automl/RoBO, (though I am sure there are newer implementations)




**Limitations:**

- as above mentioned comment, discretizations in higher dimensions are generally considered bad practice, in particular, ungioded/naive grid discretizations that include many dead points.

---

> ### Author Rebuttal · Authors · 2023-08-10
>
> We appreciate a lot for your thorough and detailed review! For the comments in "Summary",
> 1: we are really glad that you talked about the "random discretization", since we also discussed this many times when proposing the model, we will talk more about this in the following; 2: for "implementation benefits...": The idea behind is correct, although it's not Kronecker product, the mode-$k(k=1,\ldots,D$) unfolding matrix of the data tensor can be directly reconstructed with Khatri–Rao product and matrix multiplication of latent matrices. With CP decomposition we have $\boldsymbol{Y}_{(k)}=\boldsymbol{G}_k\left(\boldsymbol{G}_D\odot\cdots\odot\boldsymbol{G}_k+1\odot\boldsymbol{G}_k-1\odot\cdots\boldsymbol{G}_1\right)^{\top}$. Yes, this is the reason we reconstruct the whole grid space to select the next query point instead of random sample in the first place.
>
> For the comments in "Strengths", we are very glad that you like the idea of the model, and think it could be a universal approximator in theory. About "...discontinuities and kinks...": BKTF can model discontinuous/categorical processes through a Wishart prior on the corresponding dimensions; we mentioned this in the end of the paper, see line 364-365. In addition, BKTF (for nonstationary and nonseparable processes) could be way more flexible than a GP with SE-ARD kernel (stationary and separable). Please refer to the related discussion in the response to all.
>
> For other comments, there are mainly two concerns: (a) should consider random discretization; (b) should compare with additive GP and other baselines. We thank you for the constructive comments. We truly hope the answers below will solve your concerns and hope you can consider increasing the score for this work.
>
> $\textbf{for Weaknesses: Technical:}$
> 1. We have discussed the computational cost of BKTF, including the AF computation with grid and random discretization in the response to all. The enumeration-based AF requires more time and memory costs but also finds the optimum with fewer "experiment" budgets, especially for low dimensional problems, so it is still suggested when the AF cost is trivial compared with the problem itself. The random selection-based AF provides an alternative solution to deal with higher dimensional functions and can alleviate the curse of dimensionality, but needs more iterations of function evaluations to find the global optima. We have compared BKTF-grid (reconstruct the grid space for AF) and BKTF-random (randomly selected points for AF) on benchmark functions, the results in Table A and Figure A (see PDF) are consistent with the conclusion.
> 2. We have added additive GP, with the sum of two 1st-order additive kernel, i.e., sum of two kernel functions per dimension, as the baseline for experiments on benchmark functions. Following the reviewer's description, such additive GP matches $R=2$ BKTF. We have to say that BKTF is not equivalent to additive GP. The intrinsic difference is that the kernel representation of BKTF is built on the latent factors which are also learned from the data, not the kernel functions (see Eq.(10) in the paper). The results on the benchmark functions have also demonstrated that BKTF could be much more flexible compared with additive GP.
> 3. Thank the reviewer, we agree we should discuss more related work about the studies in BO. We have cited and discussed the recommended references in the revised paper.
>
> $\textbf{for Weaknesses: Presentation:}$
> 1. The CP decomposition is different from SVD as it does not require the basis to be orthonormal. We may say that CP decomposition is a tensor generalization of matrix factorization (MF).
> 2. Yes we definitely can.
> 3. Sorry we removed the definition here and introduced it in the Appendix before it is used.
> 4. We agree that the grid is not necessary for model inference, but Eqs. (7) and (8) are actually obtained based on the underlying model assumption of kernelized CP decomposition.
> 5. We agree that the discussion about kernel representation and connection with other GP-related models is not the main problem in the BO task, but the flexible kernel representation of BKTF is actually the key motivation of introducing the model. We will put it after Section 3.4 of the original paper to make the methodology part unbroken and easy to read.
> 6. We define the AF as Bayesian-UCB because the mean and uncertainty are obtained in a Bayesian way from MCMC samples, thus the AF does not have analytical equations for BKTF, which is different from the closed-form results in GP surrogates.
>
> $\textbf{for Questions:}$
> 1. The grid is not required for model inference and is only needed for AF computation to determine the next query point. We have discussed the computational cost of BKTF with grid and random selection-based AF strategies. We have added BKTF with random as the reviewer suggested, denoted as BKTF-random, the results on benchmark functions are given in Table A and Figure A in PDF.
> 2. Following the suggestion, we have added additive GP with the sum of two 1st-order additive kernels as the baseline in the experiments, see Table A and Figure A.
> 3. We have considered Bayesian neural network-based approaches as the baselines. Given the time limit, we do not finish the comparison experiments right now. Nevertheless, based on the results from the RoBO paper as you suggested, see ANN/NGTO and RF results in Table A, Bayesian neural networks and other non-GP approaches perform worse than GP surrogates in low dimensional problems.
>
> $\textbf{for Limitations:}$ As mentioned, we have discussed the effects of the grid for this work as much as we can.

---

> ### Comment · Reviewer_1Atp · 2023-08-17
> **Thank for the thoughtful response**
>
> I am happy to hear about the extra baselines and the discussion of practicalities do make the paper stronger, although I cannot see any results if the author would be able to share a plot or results table in this discussion I would be happy to increase my score.

---

> > ### Author Response · Authors · 2023-08-17
> > **Thank you and updated results**
> >
> > Thank you for your reply! Sure. We attached a one-page pdf in the global response (should be visible now), where a ***figure*** and a ***table*** for the updated results are included. The table is also given below:
> > | Function ($D$) | GP $\alpha_{\text{EI}}$ | GP $\alpha_{\text{UCB}}$ | GPgrid $\alpha_{\text{EI}}$ | GPgrid $\alpha_{\text{UCB}}$ | additive GP   | ANN/DNGO                    | RF            | BKTF-random            | BKTF-grid              |
> > |----------------|-------------------------|--------------------------|-----------------------------|------------------------------|---------------|-----------------------------|---------------|------------------------|------------------------|
> > | Branin (2)     | 0.01$\pm$0.01           | 0.01$\pm$0.01            | 0.31$\pm$0.62               | 0.24$\pm$0.64                | 0.05$\pm$0.09 | $\approx$0.025/$\approx$0.4 | $\approx$0.99 | $\textbf{0.00}\pm$0.00 | $\textbf{0.00}\pm$0.00 |
> > |                | $\approx$44             | $\approx$42              | $\approx$23                 | $\approx$36                  | $\approx$100  | $\approx$100/$>$100         | $>$200        | $\approx$47            | $\approx\textbf{4}$    |
> > | Damavandi (2)  | 2.00$\pm$0.00           | 2.00$\pm$0.00            | 1.60$\pm$0.80               | 2.00$\pm$0.00                | 2.00$\pm$0.00 | -                           | -             | 0.60$\pm$0.92        | $\textbf{0.00}\pm$0.00 |
> > |                | -                  | -                   | -                      | -                            | -             | -                           | -             | $\approx$48            | $\approx\textbf{5}$    |
> > | Schaffer (2)   | 0.02$\pm$0.02           | 0.02$\pm$0.02            | 0.10$\pm$0.15               | 0.09$\pm$0.07                | 0.03$\pm$0.03 | -                           | -             | $\textbf{0.00}\pm$0.00 | $\textbf{0.00}\pm$0.00 |
> > |                | $\approx$36             | $\approx$44              | $>$50                       | $>$50                        | $\approx$43   | -                           | -             | $\approx$54            | $\approx\textbf{22}$   |
> > | Griewank (3)   | 0.14$\pm$0.14           | 0.25$\pm$0.10            | 0.23$\pm$0.13               | 0.22$\pm$0.12                | 0.10$\pm$0.09 | -                           | -             | $\textbf{0.00}\pm$0.00 | $\textbf{0.00}\pm$0.00 |
> > |                | $>$100                  | $>$100                   | $>$100                      | $>$100                       | $\approx$100  | -                           | -             | $\approx$47            | $\approx\textbf{43}$   |
> > | Griewank (4)   | 0.10$\pm$0.07           | 0.19$\pm$0.12            | 0.38$\pm$0.19               | 0.27$\pm$0.17                | 0.13$\pm$0.11 | -                           | -             | $\textbf{0.00}\pm$0.00 | $\textbf{0.00}\pm$0.00 |
> > |                | $>$100                  | $>$100                   | $>$100                      | $>$100                       | $>$100        | -                           | -             | $\approx$87   | $\approx\textbf{68}$   |
> > | Hartmann (6)   | 0.12$\pm$0.07           | 0.07$\pm$0.07            | 0.70$\pm$0.70               | 0.79$\pm$0.61                | 0.48$\pm$0.17 | $\approx$0.14/$\approx$0.21 | $\approx$0.52 | 1.41e-5$\pm$1.73e-5  | $\textbf{0.00}\pm$0.00 |
> > |                | $>$100                  | $>$100                   | $>$100                      | $>$100                       | $>$100        | $>$200/$\approx$200         | $>$200        | $\approx$154           | $\approx\textbf{60}$   |
> > | Griewank (10)  | 0.36$\pm$0.07           | 0.38$\pm$0.10            | -                           | -                            | 0.25$\pm$0.30 | -                           | -             | $\textbf{0.00}\pm$0.00 | -                      |
> > |                | $>$200                  | $>$200                   | -                           | -                            | $>$150        | -                           | -             | $\approx\textbf{124}$  | -                      |
> >
> > The table compares optimization performance on benchmark functions, where ***First row***: $\left|f^{\star}-\hat{f}^{\star}\right|$ when $n=N$ (mean$\pm$std.); ***Second row***: Average costs for the number of evaluations to find the optimum. The results from more baselines and on 10D Griewank function are given. The results of ANN/DNGO (Bayesian neural networks related methods) and RF(random forest) are from Figure 3.3 in the reference paper [R1].
> >
> > A ***figure*** showing the detailed optimization results is provided in the ***pdf*** attached in the global response (response to all), please download the pdf file and refer to ***Figure A*** for more results. We will add the baselines and results in the revised paper, thank you again for your comments.
> >
> > [R1]: A. Klein, “Efficient bayesian hyperparameter optimization,” Ph.D. dissertation, Dissertation, Universit ̈at Freiburg, 2020.

---

### Official Review · Reviewer_BFf9 · 2023-06-29

**Soundness:** 3 good
**Presentation:** 3 good
**Contribution:** 3 good
**Rating:** 7
**Confidence:** 3

**Summary:**

The paper presents a new surrogate model called Bayesian Kernelized Tensor Factorization (BKTF) for Bayesian Optimization (BO).The BKTF model approximates the solid in the D-dimensional space using a fully Bayesian low-rank tensor CP decomposition. It uses Gaussian process (GP) priors on the latent basis functions for each dimension to capture local consistency and smoothness. This formulation allows sharing of information not only among neighboring samples but also across dimensions. The paper proposes using Markov chain Monte Carlo (MCMC) to efficiently approximate the posterior distribution. ). The paper demonstrates the effectiveness of BKTF through numerical experiments on standard BO test functions and machine learning hyperparameter tuning problems.

**Strengths:**

1. One of the significant strengths of the paper is the novel and reasonable solution of incorporating the idea of tensor decomposition into Bayesian Optimization (BO). This approach allows for a more efficient and effective representation of the D-dimensional Cartesian product space, enhancing the performance of BO. The adoption of tensor decomposition represents a significant advancement in the field and demonstrates the authors' innovative thinking.

2. The usage of two-layer GPs is impressive. This approach is clever as it allows for the sharing of information among neighboring samples and across dimensions, enhancing the model's ability to capture local consistency and smoothness.



**Weaknesses:**

1. The cost of several cascaded full GPs may be high, especially for cases with large nodes (refer to "coordinate" in paper) at some dimension. More discussions are encouraged on the scalability analysis or the possible solutions, such as sparse GP,  to reduce the cost.

2. As the tensor rank R  is always a crucial hyperparameter for tensor decomposition. I'm curious about how the rank setting could influence the BO. It will be great if the authors could give some comments or results on why R=2 is sufficient for the model setting.

**Questions:**

See Weakness

**Limitations:**

See Weakness

---

> ### Author Rebuttal · Authors · 2023-08-09
>
> Thank you for your positive and detailed review! We are very glad and appreciate that you agree with the novelty of this work. We explain the weaknesses below, and hope the answers will address your concerns.
> 1. For the model scalability, we have discussed the computational cost of BKTF in terms of model inference and AF computation in the response to all. As we mentioned, the time cost of model inference for BKTF should be the same as GP inference ($\mathcal{O}\left(n^3\right)$, $n$ is the number of observable data points) if using point-wise sampling, and could be better than GP when $n$ becomes larger by utilizing the low-rank factorization, which is $\mathcal{O}\left(\sum_{d=1}^{D}|S_d|^3\right)$. For data off the grid, we can indeed leverage sparse GP and inducing points/grid for fast inference; similar to the idea used in KISS-GP. Nevertheless, for BKTF, we would like to highlight that the main cost is caused by enumeration-based AF computation when selecting the next query point, which increases exponentially with the dimensions. We believe this is an important follow-up research question. A possible solution is to randomly select certain candidate points in the defined space for AF instead of reconstructing the whole data tensor. For the experiments in the current paper, BKTF with grid-based AF query strategy is acceptable, we will give the running time per function evaluation of different models on the tested benchmark functions later. We have also evaluated a higher dimensional 10D benchmark function and considered random discretization for AF of BKTF, denoted as BKTF-random (see the updated results in Table A and Figure A in PDF). The results show that BKTF with randomly selected AF provides an alternative model to handle higher dimensional problems that BKTF-grid is not feasible, and it still obtains the best optimization performance compared with other models. This demonstrates the superior advantage of the underlying BKTF framework.
> 2. For the rank setting, the reason we choose a small rank i.e., $R=2$ is that the number of available/observable data is very small, e.g., less than 100. In such cases, a small rank is enough to capture the flexibility/correlations of the data and can estimate a surface with adequate uncertainty quantification through highly efficient inference. In addition, since the task in BO is to find the global optimal values with the smallest budget, the accuracy of the reconstructed surface with respect to the true surface is actually not the primary goal. For problems with a large number of observations, one can use a large $R$ or extend BKTF to a non-parametric Bayesian version to automatically learn/adjust rank $R$ based on the data.

---

### Official Review · Reviewer_LVy9 · 2023-07-05

**Soundness:** 3 good
**Presentation:** 3 good
**Contribution:** 3 good
**Rating:** 7
**Confidence:** 4

**Summary:**

This paper presents a surrogate based on tensor decompositions for approximating complex functions, allowing for Bayesian-style maximization.
Numerical experiments show the (slight) superiority of this model over classical Bayesian approaches. However, a limitation of this approach is the small dimensionality of the target functions and the need to use a discrete grid.



**Strengths:**


- The proposed algorithm uses a very small budget to find the maximum of complex functions (gradient-free, multimodal).

- A good potential for expanding and improving the proposed algorithm.

- This article uses a tensor approach for machine learning problems

- The presented new algorithm, in my opinion, has quite a lot of possibilities for improvement, and the article itself is complete.

**Weaknesses:**

- A small number of numerical examples.

- Final accuracy in Fig. 3b better, but very close to the accuracy of the other methods with which the comparison is made.

- No comparison with non-Bayesian methods of finding the maximum.

**Questions:**

In the line 96 of the text, a random noise $\epsilon_i$ is added to the real values of the approximating function. Did you add the noise during numerical experiments to the model functions (Branin, Schaffer, Griewank, etc). If so, what was the variance of the noise?

Did you try to run you algorithm on higher dimension (with less $m_d$ values), or more complex functions which need more budget for convergence?

What is the characteristic running time of the proposed algorithm compared to the others mentioned in the article?




**Limitations:**

Small dimension of functions for which the maximum is searched for. This is due to the fact that AF has to be found by unrolling the tensor from CP to the full format.
Thus, one of the main advantages of the CP tensor format, related to overcoming the curse of dimensionality, is not used.

---

> ### Author Rebuttal · Authors · 2023-08-09
>
> Thank you for your review! We are really glad that you agree with the potential of this work. The main concern is the small dimensionality of the test functions, i.e., the scalability of the model, which relates to the computational cost and grid assumption. $\textbf{We have explained the computational cost in the response to all}$, and $\textbf{conducted more experiments with more baselines and discussed the running time.}$ We explain the weaknesses,  answer the questions, and reply to the limitations one by one below.
>
> $\textbf{for Weaknesses:}$
> 1. We conducted more experiments. The updated results on benchmark functions are given in Table A and Figure A (see PDF), where a higher dimension 10D Griewank function is tested, and more baselines are considered. Given the time limit, we only give the results on one more function, but we are also testing on more complicated but low dimensional functions (Multi-modal functions with global structure: the Rastrigin Function and Weierstrass Function) and extending the experiments in Section 5.2 (adding baselines and categorical inputs) as well. We believe the conclusions are similar: the proposed BKTF surrogate can achieve superior performance for moderate dimensional BO tasks, particularly with severely limited budgets; we will update the experiments in the later revised paper.
> 2. We think the reviewer means the classification accuracy results in Figure 3(a), where BKTF (blue lines) is faster to find better hyperparameter combinations for the algorithms on the MNIST dataset. The final accuracy is close because the MNIST dataset we used (1797 samples) is simple and 0.2\% improvement in accuracy is sufficiently better.
> 3. Actually the GP surrogate-based baselines: i.e., GP/GPgrid $\alpha_{\text{EI}}$, GP/GPgrid $\alpha_{\text{UCB}}$, are not Bayesian methods, since the GP surrogates have analytical solutions for the uncertainty and mean, i.e., acquisition functions (AF), and can be directly applied for BO tasks. We have added an additive GP surrogate with EI as a new baseline model, which is not Bayesian either. If the reviewer is talking about non-GP comparison models, we can look into the work in [R1] (Figure 2) and [R2] (Figure 3.3), in which Bayesian neural network-based BO approaches (e.g., BOHamiANN) and other non-GP-methods-such-as random forest (RF) are also compared on the low dimensional benchmark functions, e.g., 2D Branin Function (also tested in this paper). From the results figures, they clearly said that GP surrogates obtain the best performance among the compared models for low dimensional function optimization.
>
> $\textbf{for Questions:}$
> 1. We did not add the noise to the model functions. The random noise $\epsilon_i$ is added only for estimation. But it is straightforward to add white noise to the objective model functions since this does not affect the estimation process. In estimation, we assume the data is noisy with a white noise no matter how the real/true data is generated.
> 2. Yes. We added the experiments on a 10D benchmark function, and are still conducting several other experiments on more complex low-dimensional benchmark functions and extending the experiments in Section 5.2. From the current updated results, we see that clearly the proposed BKTF consistently provides the best performance.
> 3. We have discussed the computational cost in the response to all. Compared to the GP surrogate baselines, the cost of model inference should be similar when using the point-wise updating strategy, which is $\mathcal{O}\left(n^3\right)$, $n$ is the number of observation points. As for the time cost of AF computation, as we explained in the response to all, when the cost of each BO iteration is trivial compared with the cost of the design experiment itself, the full enumeration (as used in our paper) is acceptable since it provides more efficient solution to the optimization problems. When the cost of BO becomes considerable compared with the design experiment, one should develop more efficient solutions for AF computation, such as using random (see the new BKTF-random results) or local subsets as suggested by the reviewer. For the experiments conducted in the paper, we think the running time per iteration of evaluation for BKTF is acceptable and is similar to the baseline models in low-dimensional problems. Thus the total running time of BKTF for finding the global optima should be better than others since it costs the least number of iterations. Given the time limit, we will give the average running time of different models on the benchmark functions later in the paper.
>
> $\textbf{for Limitations:}$ Yes the unrolling of the tensor for AF computation has the scalability issue. We believe this is an important follow-up question to answer when extending the proposed framework for problems with higher dimensionality. Right now we have only evaluated the alternative of using a random subset for AF for higher dimensional problems, see BKTF-random. But we believe the underlying advantage/contribution of BKTF is the same, that is introducing a more flexible and elegant fully Bayesian surrogate that can capture/leverage the global correlations and obtain high-quality uncertainty quantification from the limited data to achieve efficient global search with limited budgets.
>
> [R1] A. Klein, S. Falkner, N. Mansur, and F. Hutter, “Robo: A flexible and robust bayesian optimization framework in python,” in NIPS 2017 Bayesian optimization workshop, 2017, pp. 4–9.
>
> [R2] A. Klein, “Efficient bayesian hyperparameter optimization,” Ph.D. dissertation, Dissertation, Universit ̈at Freiburg, 2020.

---

> > ### Comment · Reviewer_LVy9 · 2023-08-16
> >
> > Thank you for the very detailed response. After reading your answer and discussion with other reviewers ,I am going to keep the positive score 7: Accept.

---

### Author Rebuttal · Authors · 2023-08-09

Dear reviewers,

Thank you for your time and the thorough and invaluable feedback. We appreciate that the contribution of our paper is well perceived and recognized, as reflected in the contribution scores (3, 3, 3, 3). We are also pleased that the concept of introducing a kernelized low-rank factorization model as a surrogate for Bayesian Optimization (BO) resonates with you. In this general response, we would like to highlight two primary areas that have garnered your attention: (a) numerical experiments are not enough, in terms of the tested functions and baseline models; (b) computational cost of the model, related to grid assumption and the scalability issue. We are conducting more experiments on higher dimensional benchmark functions with more comprehensive baselines. However, given the time limit, several experiments are still ongoing and we will update the results in the later version of the rebuttal.

Below we will address and clarify some common points from the reviewers.
1. The definition of $\textbf{nonstationary}$ and $\textbf{nonseparable}$ processes. We should provide a brief introduction in the main paper. A stationary covariance function depends only on the distance of the data points and should be invariant to the specific locations. The commonly used SE kernel is stationary as it is determined by $|x-x'|$; a linear kernel is not stationary as the covariance value is location-specific. For additive GP, if all the component kernels are stationary, the final kernel function will still be stationary. A covariance/kernel function is separable if $k\left(\boldsymbol{x},\boldsymbol{x}'\right)=k_1\left(x_1,x_1'\right)k_2\left(x_2,x_2'\right)\cdots k_D\left(x_D,x_D'\right)$, thus implying the independence between the input dimensions. The commonly used SE-ARD kernel is a separable kernel. Stationary and separable kernel functions offer computational advantages; however, they are limited in modeling functions/processes with complex dependency structures.

2. The $\textbf{computational cost}$ of the proposed BKTF surrogate.
For the cost of model inference, when the number of observations $n$ is small, we use point-wise sampling, and the computational cost is the same as a full GP $\mathcal{O}\left(n^3\right)$. When $n$ becomes larger, we can update the model based on the latent factors leveraging the low-rank structure, the computational cost is $\mathcal{O}\left(\sum_{d=1}^D|S_d|^3\right)$, which could be more efficient than $\mathcal{O}\left(n^3\right)$. The cost for model inference should be at least better than GP inference. The main cost comes from the computation of AF. To select the next query point, in the current paper we simply unroll the whole data tensor in the defined grid space (i.e., enumeration-based AF) and the costs increase exponentially with the dimensions. A potential solution (as mentioned by the reviewer) is to randomly select candidate points or develop more efficient strategies instead of reconstructing the whole space. In this way, this can alleviate the curse of dimensionality but at the cost of more evaluation budget (BO iterations). We test random discretization/selection, denoted as BKTF-random, on benchmark functions and a 10D Griewank function. The results are consistent with our assumptions: BKTF-random can be applied for higher dimensional problems that cannot be performed with a grid but costs more iterations for low dimensional functions compared with BKTF-grid. When the cost of the experiment itself is highly expensive (e.g., some real-world optimization problems may need multiple hours or days to solve), an additional overhead from the AF computation of a few seconds or minutes is acceptable if a better optimization efficiency is achieved. But when the cost of the AF is not acceptable, we would suggest using more efficient searching strategies. More solid research is also needed to design an AF for BKTF that balances the two construction strategies; we take this as an important future research question.

3. For the $\textbf{experiments}$, we test on a higher dimension 10D Griewank function and are testing other low dimension but complex functions (Rastrigin and Weierstrass Function), and extending the experiments in Section 5.2 as well. In terms of the $\textbf{baseline models}$: we have added: (a) additive GP with two 1st-order additive kernels per dimension (same number of latent functions as BKTF but in a sum-based manner); (b) BKTF with random discretization; (c) non-GP approaches, such as Bayesian neural networks and RF. The updated results are given in Table A and Figure A in PDF; more results will be updated soon.

Overall, we believe BKTF provides an elegant framework that can achieve more efficient and stable performance for BO tasks particularly with severely limited budgets. We hope the response can address your concerns.

---

### Decision · Program_Chairs · 2023-09-21

**Decision:**

Reject

**Comment:**

This paper presents a new surrogate model, 'Bayesian kernelized Tensor Factorization' for Bayesian optimization. The main idea is to approximate a complex function with a Bayesian low-rank CP decomposition, which is a continuous analogue of the eigen-decomposition of a matrix. A downside in this approach is the inference is not of the analytic form, in contrast to the GP. Most of reviewer feel that the paper has some interesting contributions. In addition, the authors did a good to resolve most of concerns raised by reviewers. However, some concerns remained even after the author rebuttal. In particular, one reviewer feels that the baselines are not in the highest standards for this line of work. More limitations can be found in the comments. Therefore, the paper is not recommended for acceptance in its current form. I hope authors found the review comments informative and can improve their paper by addressing these carefully in future submissions.